# Circulation of *Herpesvirus* and *Alphatorquevirus* DNA in each trimester in asymptomatic women pregnant with twins

Tania Regina Tozetto-Mendoza[1]*, Layla Honorato[1], Mariane Pereira Brito[1], Noely Evangelista Ferreira[1], Iara M. Linhares[2], Angela Silvano[3], Antonio Charlys da Costa[1], Silvia Helena Lima[1], Viola Seravalli[3], Mariarosaria Di Tommaso[3‡], Maria Cassia Mendes-Correa[1‡], Steven S. Witkin[1,4‡]

1 Laboratory of Virology (LIM 52), Department of Infectious Diseases and Tropical Medicine, Instituto de Medicina Tropical de São Paulo, Medical School, University of São Paulo, São Paulo, Brazil, 2 Department of Gynecology and Obstetrics, University of São Paulo Medical School, São Paulo, Brazil, 3 Division of Obstetrics and Gynecology, Department of Health Sciences, Careggi Hospital, University of Florence, Florence, Italy, 4 Department of Obstetrics and Gynecology, Weill Cornell Medicine, New York, New York, United States of America

‡ These authors jointly supervised this work.
* tozetto@usp.br

## Abstract

### Background

Data on the dynamics of Herpesviruses and Torque teno virus (TTV) in plasma from twin pregnancies are limited, despite their potential to provide insights into maternal immune status in this high-risk population.

### Methods

A convenience sample of plasma was obtained from 54 healthy women with twin pregnancies and no adverse outcomes. The plasma samples were collected at 8–15, 17–25, and 26–35 weeks of gestation at Careggi University Hospital, Florence, Italy. Real-time PCR detected Herpesviruses and TTV DNA, determined viral titers, and evaluated associations with cytokine levels and pregnancy-related parameters.

### Results

Herpesvirus were detected in 10 women (18.5%): three for herpes simplex virus (HSV)-1, three for human herpesvirus (HHV)-6, one for both HSV-1 and HHV-6, and one each for HSV-2, cytomegalovirus (CMV), or Epstein–Barr virus (EBV). Only one woman was positive throughout all trimesters (HHV-6). TTV DNA was detected in 42 women (77.7%), persistently in 26 (48.1%) and transiently in 16 (29.6%). No associations were found between viral detection and plasma interleukin (IL)-1β, IL-6, tumor necrosis factor-α (TNF-α), or pregnancy-related parameters, except for a higher prevalence of TTV in women with previous deliveries (p = 0.0402).

**Data availability statement:** All relevant data are within the manuscript and its Supporting information files.

**Funding:** The support for the laboratorial investigation was provided by the Laboratory of Medical Investigation (LIM) 52, Faculty of Medicine, University of São Paulo, Brazil.

**Competing interests:** The authors declare that they have no known competing financial interests or personal relationships that could have appeared to influence the work reported in this paper.

## Conclusion

The high frequency and variable viral titer of TTV with the transient circulation of some herpesviruses appear unaffected by the immunological adaptations inherent to this setting of twin pregnancies without adverse outcomes, as supported by the absence of correlation between inflammatory markers and viral dynamics. Nonetheless, ongoing monitoring of obstetric and postnatal outcomes remains warranted.

## Introduction

Torque teno virus (TTV) and herpesviruses DNA in maternal plasma from twin pregnancies remain largely unexplored, despite their potential to reveal key aspects of maternal immune status.

TTV is an *Alphatorquevirus* and belongs to the *Anelloviridae* family. It is often considered nonpathogenic and is detectable in multiple body fluids [1]. Its replication is thought to occur in activated T lymphocytes [2], and fluctuations in its plasma have been proposed as a biomarker of immune competence [3], particularly in the context of immunosuppression following transplantation [4,5] or during chronic infections [6,7]. In pregnancy, few studies have addressed the role or clinical relevance of TTV [8–12]. In general, TTV is a prevalent commensal virus in pregnant women, with its detection and titer reflecting local and systemic immune status, vaginal microbiome composition, and possibly reproductive outcomes. Its increased presence and titer may serve as a potential surrogate marker for immune status, CD4 level in variable clinical conditions [11–14]. A recent investigation reported a high prevalence of TTV in maternal plasma, with detection rates exceeding 75% during the second and third trimesters [9,10]. Additionally, other studies have linked vaginal TTV levels to bacterial composition and mucosal immune activity [11,12].

Members of the *Herpesviridae* family, herpes simplex virus (HSV) −1 and −2, cytomegalovirus (CMV), human herpes virus (HHV)-6, −7 and −8, Epstein Barr virus (EBV) and varicella zoster virus (VZV)—establish lifelong latency following primary infection [15]. Under conditions of immune modulation, these viruses may intermittently reactivate and replicate their DNA, particularly HSV-1 and CMV, which are strongly associated with perinatal or congenital infection and an increased risk of adverse pregnancy outcomes [16–20].

Twin pregnancies present a distinct immunological milieu compared to singleton gestations, with increased antigenic load and greater placental mass, requiring finely tuned adaptations to support fetal tolerance, placental development and protection against infection. On the other hand, twin gestations may increase susceptibility to immune-mediated complications and alter responses to viral infections [21–23]. Additionally, measurable immunological changes during gestational progression, such as elevated circulating concentrations of IL-6, TNF-α, and IL-1β across all trimesters, have been reported in twin pregnancies [24]. Given these immunological complexities, and the fact that the prevalence of twin pregnancies has steadily increased in

recent decades due to assisted reproductive technologies and delayed maternal childbearing, this population represents a previously rare and understudied setting.

To our knowledge, no study has longitudinally assessed the presence of circulating TTV and herpesviruses across all three trimesters in women with uncomplicated twin pregnancies. This study provides the first longitudinal quantification of TTV and herpesvirus in maternal plasma and examines their inflammatory correlates, offering novel insights into viral dynamics during twin pregnancies.

## Materials and methods

### 1.1. Women with twin pregnancies

Participants in this investigation were women pregnant with twins who were seen at Careggi University Hospital in Florence, Italy between January 2018 and February 2022. Exclusion criteria included a symptomatic maternal infection as determined by the woman's clinician, presence of a fetal anomaly, pregestational diabetes, use of a cervical cerclage to prevent premature delivery or loss of follow-up from antenatal care.

This study included a convenience sample of 54 asymptomatic healthy women without adverse clinical outcomes, with plasma samples available across all three trimesters of pregnancy, including both spontaneous and ART-conceived pregnancies, as well as dichorionic and monochorionic pregnancies. In our cohort, there were no cases of chronic comorbidities or pregestational diabetes, and no cases of twin-to-twin transfusion syndrome (TTTS) were observed. Moreover, there were no suspected cases of maternal infection, as determined by the absence of clinical symptoms in mothers or newborns. The median age at sampling was 35 years (IQR: 33–38), with values ranging from 26 to 46 years.

Gestational age in spontaneous conceptions was calculated from the last menstrual period and confirmed by the fetal crown-rump length measurement from a first trimester ultrasound. In pregnancies conceived by assisted reproductive technologies (ART) embryonic age was determined from the time of fertilization. Reasons for ART included occluded Fallopian tubes, anovulation or inadequate sperm parameters in the male partner. ART included *in vitro* fertilization or intracytoplasmic sperm injection utilizing either autologous or donor oocytes.

Blood samples were collected from each woman at three time points: 8−15 weeks, 17−26 weeks and 27−35 weeks of gestation. Blood processing and measurement for the pro-inflammatory cytokines interleukin (IL) −1β, IL-6 and tumor necrosis factor-α in these women have been previously described in twin pregnant women and showed significant increase levels across the trimesters of gestations [24].

All women provided written informed consent before inclusion. The study was approved by the Ethical Committee of Azienda Ospedaliero-Universitaria Careggi (Ref. no.10255/2017).

### 1.2. DNA extraction and quality control

Total nucleic acids were extracted and purified from 200 µL of plasma using a magnetic bead- based automated system (Extracta® Kit – DNA and RNA Viral, MVXA-PV96-B FAST, Loccus, Cotia, São Paulo, Brazil) on the Extract platform (Loccus, Brazil), following the manufacturer's protocol. The final elution volume of purified nucleic acids was 100 µL. DNA quality and the absence of PCR inhibitors were assessed by amplification of an internal control targeting the human Ribonuclease P gene, using primers and probe as previously described [24,25]. All samples were deemed suitable for downstream DNA analysis as evidenced by successful amplification of the control.

### 1.3. Viral detection and quantification

Quantitative real-time PCR (qPCR) assays were performed using hydrolysis probes labeled at the 5′ end with FAM, VIC, or NED reporter dyes and a 3′ MGB-NFQ quencher. Oligonucleotide primers and probes were synthesized by Integrated DNA Technologies (IDT, USA), based on previously published sequences for the genus *Alphatorquetenovirus* [26,27] and

subfamilies *Alphaherpesvirinae* [28,29], *Betaherpesvirinae* [30,31] and *Gammaherpesvirinae* [32,33]. Oligonucleotide sequences of synthetic curves for the herpesviruses quantitative real-time PCR are delineated in supplementary Table S1 in S1 File.

Reactions were performed using TaqMan® Universal PCR Master Mix (Thermo Fisher Scientific, Warrington, UK) in a final volume of 25 µL, containing 100 ng of DNA template. Final concentrations were 200 nM for primers and 62 nM for probes. Amplification was carried out on a QuantStudio™ 5 Real-Time PCR System (Thermo Fisher Scientific, Warrington, UK) under the following cycling conditions: 50°C for 2 minutes, 95°C for 15 seconds, followed by 50 cycles of 95°C for 15 seconds and 60°C for 1 minute. Absolute quantification of viral DNA was achieved using a standard curve generated from synthetic DNA templates corresponding to each target viral region. The sequences of these templates were synthesized and HPLC-purified by Exxtend Biotecnologia Ltda (São Paulo, Brazil), reconstituted in TE buffer (10 mM Tris-HCl, 0.1 mM EDTA, pH 8.0) at a concentration of 100 µM and serially diluted 10-fold in nuclease-free water. Standard curve generation followed previously established protocols [27,34]. Data were analyzed using QuantStudio Design & Analysis Software v1.4.1. The lower limit of detection (LoD ≥ 95%) varied from 1.4 to 2.0 $\log_{10}$ copies/mL: TTV (1.6 $\log_{10}$ copies/mL), HSV-1 and HHV-8 (1.7 $\log_{10}$ copies/mL), HSV-2 and HHV-6 (1.4 $\log_{10}$ copies/ml), VZV and HHV-7 (1.9 $\log_{10}$ copies/mL), EBV (1.5 $\log_{10}$ copies/ml) and CMV (2.0 $\log_{10}$ copies/mL).

### 1.4. Statistics

Continuous variables were assessed for normality using the Shapiro-Wilk test. Comparisons were performed using one-way ANOVA or the Kruskal-Wallis test for multiple groups, and the Mann-Whitney test, as appropriate. Spearman's rank correlation was used to evaluate the relationship between TTV titers ($\log_{10}$ copies per ml), maternal age and the concentrations of IL-1β, IL-6, and TNF-α during each trimester of pregnancy. Categorical variables were analyzed using Fisher's exact test or the Chi-square test, as appropriate. A p-value < 0.05 was considered statistically significant. All analyses were performed using GraphPad Prism version 10 (San Diego, CA).

## Results

### 2.1. Herpesviruses and TTV DNA detection ant titer in twin pregnancies

Detection and titer of herpesviruses DNA and the number of women positive for the individual viruses in each trimester are shown in Table 1. Ten of the 54 women (18.5%) were virus DNA positive: one for both HSV-1 and HHV-6, three for only HSV-1, three for only HHV-6, and one each for only HSV-2, CMV and EBV. All samples were DNA negative for VZV and

**Table 1. Herpesviruses DNA titer in plasma of women with twin pregnancies.**

| DNA herpes | Number of positive tests | 1st Trimester n total (median titer)* | 2nd Trimester n total (median titer) | 3rd Trimester n total (median titer) |
|---|---|---|---|---|
| HSV-1 | 4 | 1 (4.12) | 1 (3.93) | 2 (3.81, 4.02) |
| HSV-2 | 1 | 0 (0) | 0 (0) | 1 (3.60) |
| VZV | 0 | 0 | 0 | 0 |
| EBV | 1 | 0 (0) | 1 (3.49) | 0 (0) |
| CMV | 1 | 0 (0) | 1 (3.42) | 0 (0) |
| HHV-6 | 7 | 4 (3.20, 3.35, 4.55, 4.61) | 2 (3.41, 4.73) | 1 (4.21) |
| HHV-7 | 0 | 0 | 0 | 0 |
| HHV-8 | 0 | 0 | 0 | 0 |

ªmedian titer in copies/ml, n = number of positive DNA tests; n = number of positive DNA tests.

HHV-8. Most of the viruses were detected only transiently during one or two of the trimesters of pregnancy. There was no apparent difference in the number of virus-positive women by trimester. Five women each were herpesvirus DNA positive in the first or the second trimester while four were positive in the third trimester. Only one woman was virus DNA positive for HHV-6 during her entire gestation. No significant differences were observed across the trimesters of twin pregnancy (Table 1). The supplementary Figure 1S in S1 File presents the titer of detected viruses across all trimesters.

The frequencies of TTV DNA were 68.5% (37 women), 59.3% (32), and 61.1% (33) in the first, second, and third trimesters, respectively (p = 0.5736). TTV DNA was detected in 42 (77.7%) of 54 women. Unlike the situation with herpesviruses, TTV was present in all three trimesters in 25 women (46.3%). Eight women (14.8%) were TTV DNA positive in only one trimester while an additional eight (14.8%) were TTV DNA positive in two trimesters. Median TTV titer was 5.29 (range: 2.58–7.53) $\log_{10}$ copies/mL. Titers did not differ significantly between trimesters of twin pregnancy (p = 0.2246).

## 2.2. Viruses DNA detection in relation to pregnancy parameters

Pregnancy-related parameters in women positive or negative for DNA of herpesvirus or TTV are presented in Table 2. There was no association between the presence of herpesviruses or TTV and any pregnancy outcome parameter, such as premature preterm rupture of membranes, preterm delivery, gestational age at delivery or birthweight of the first and second twin. Similarly, maternal historical parameters, such as maternal age, European residency, history of spontaneous or induced abortions, history of preterm birth and body mass index were similar in all groups. Lastly, developments related to the index pregnancy, such as conception by ART, use of donor oocytes, development of gestational diabetes mellitus, hypertension or anemia, were also unaffected by the presence of either virus family. Similar results were obtained when comparing only women positive for HHV-6 or HSV-1 vs. virus negative women. The only significant difference noted was

**Table 2. Characteristics of women with twin pregnancy in relation to viral status.**

| Parameter | Herpesvirus positive (n = 10) | Herpesvirus negative (n = 44) | TTV positive (n = 42) | TTV negative (n = 12) |
|---|---|---|---|---|
| Mean age (years, SD) | 34.9 (4.2) | 35.6 (4.2) | 35.3 (4.3) | 36.1 (4.1) |
| European origin | 9 (90%) | 42 (95.5%) | 38 (92.7%) | 12 (100%) |
| [a]Parity | | | | |
| 0 | 6 (60%) | 30 (68.2%) | 25 (59.5%) | 11 (91.6%) |
| >1 | 4 (40%) | 14 (31.8%) | 17 (40.5%) | 1 (8.3%)[a] |
| Spontaneous abortions | 1 (10%) | 10 (22.7%) | 10 (24.4%) | 1 (7.7%) |
| Induced abortions | 0 | 7 (15.9%) | 5 (12.2%) | 2 (15.4%) |
| Mean BMI (kg/m², SD) | 21.5 (2.8) | 20.9 (4.8) | 21.0 (4.0) | 21.2 (5.5) |
| ART conception | 3 (30%) | 15 (34.1%) | 13 (31.7%) | 5 (38.5%) |
| Donor oocyte | 2 (20%) | 4 (9.1%) | 4 (9.8%) | 2 (15.4%) |
| Previous PTB | 0 | 0 | 0 | 0 |
| Anemia | 2 (20%) | 16 (36.4%) | 15 (36.6%) | 3 (23.1%) |
| Pregnancy hypertension | 1 (10%) | 1 (2.3%) | 2 (4.9%) | 0 |
| GDM | 2 (20%) | 13 (29.5%) | 14 (34.1%) | 1 (7.7%) |
| PPROM | 2 (20%) | 5 (11.4%) | 5 (12.2%) | 2 (15.4%) |
| Mean GA at delivery (wks, SD) | 35.8 (1.1) | 36.1 (1.6) | 36.0 (1.4) | 36.0 (1.9) |
| Preterm birth <37 wks | 8 (80%) | 26 (59.1%) | 27 (65.9%) | 7 (53.8%) |
| Mean birthweight 1st twin (g, SD) | 2271 (306) | 2287 (396) | 2311 (343) | 2201 (483) |
| Mean birthweight 2nd twin (g, SD) | 2283 (338) | 2393 (585) | 2372 (375) | 2376 (398) |

BMI, body mass index; ART, assisted reproductive technology; PTB, preterm birth; GDM, gestational diabetes mellitus; PPROM, preterm premature rupture of membranes; GA, gestational age; bwt, birthweight; wks, weeks. [a]p = 0.0438 vs TTV positive.

that women positive for TTV were more likely to have had a previous delivery as opposed to women negative for this virus (p = 0.0438). Maternal age did not show a significant correlation with TTV viral load across the three trimesters (Spearman r = −0.021, 0.020, and −0.115 for the first, second, and third trimesters, respectively; 95% CI: −0.2971 to 0.2584, −0.2593 to 0.2963, −0.3811 to 0.1678; p = 0.882, 0.887, 0.411).

### 2.3. Cytokines and viruses DNA detection

Data on cytokine level in each trimester in relation to the viral load and virus-positive or negative for herpesviruses or TTV DNA are shown in Supplementary tables (Tables S1 and S2 in S1 File). Plasma concentrations of IL-1β, TNF-α and IL-6 did not differ between women testing herpesvirus-positive or virus-negative DNA in any trimester (Table S1 in S1 File), as well as there was no statistically significant correlation between TTV titer and cytokines level in any trimester (Table S3 in S1 File). Comparable findings were observed when restricting the analysis to HHV-6- or HSV-1-positive women.

### Discussion

The present investigation, conducted on a rare cohort of twin pregnancies without adverse clinical outcomes, documents longitudinal evidence of the transient presence of herpesviruses and TTV DNA in maternal plasma. Our findings demonstrate that these viral dynamics occur independently of cytokine fluctuations across trimesters in women with twin gestations. This observation extends and reinforces previous reports in singleton pregnancies [9,35,36].

The interpretation of herpesvirus DNA detection in maternal plasma remains complex, as these viruses establish lifelong latency and can reactivate under various physiological conditions, especially during periods of hormonal and immunological shifts, such as those occurring in pregnancy [37]. Previous studies reported herpesvirus DNA can be intermittently detected in plasma from pregnant women without clinical consequences [35,36,38]. While the presence of this viral DNA does not definitively indicate ongoing lytic replication, it may reflect low-level viral activity, transient shedding from a latent reservoir, or even residual DNA from a past lytic event. Regardless of the precise biological interpretation, the detection of these viruses in plasma is clinically important during pregnancy, as it demands a careful assessment of potential vertical transmission risks and suggests that sustained maternal and neonatal monitoring is warranted.

Although our cohort was composed of healthy pregnant women without adverse clinical outcomes, it is crucial to recognize that the intermittent replication of viruses such as HSV-1 and CMV, for example, has been strongly correlated with an increased risk of perinatal or congenital infections [18–20]. In addition, twin gestations impose distinct immunological challenges and may be associated with a potential risk of adverse outcomes compared to singleton gestations [39].

In this context, HHV-6 DNA was the most frequently detected herpesvirus (in variable titers) in these pregnant women. Only one woman in our cohort was persistently HHV-6 DNA-positive across all three trimesters. She was a 36-year-old, white woman of European origin with a body mass index of 21.5, no prior pregnancies, and a spontaneous conception resulting in the delivery of healthy female and male infants at 37.1 weeks gestation. HHV-6 DNA was detected in 7.4% of this cohort of twin pregnancies without any adverse clinical outcomes, a proportion exceeding the 0.4–2.9% reported in the general population [40]. Although this finding suggests that HHV-6 detection does not invariably lead to adverse events, the potential for reactivation or congenital infection, including cases associated with inherited chromosomally integrated HHV-6, warrants attention given previously reported risks [9,38,41–46].

TTV DNA was frequently detected in maternal plasma across all trimesters of twin pregnancies,
consistent with its ubiquitous distribution [10,47]. No correlation was observed between TTV or herpesvirus DNA dynamics and pro-inflammatory plasma markers, likely reflecting the distinct clinical status of healthy women pregnant with twins who delivered at term and the exclusion of women with detectable adverse health parameters. Nonetheless, TTV remains of interest as a potential marker of local immune status of pregnancy [15], preterm birth [38] and labor-related events [10,47].

For instance, a significantly higher detection of anelloviruses in maternal serum of women experiencing spontaneous labor has been reported [10,47], suggesting a potential role in labor-associated inflammatory signaling. Similarly, TTV has been identified as the predominant component of the maternal plasma virome and an association with preterm birth risk has been reported [38].

In addition, the presence of TTV DNA in the maternal circulation was significantly more frequent in women with a history of prior pregnancy and delivery as opposed to women negative for this virus. The composition of the vaginal microbiome in women with a prior pregnancy differs from that of nulliparous women [28] and this likely is a consequence of pregnancy-induced immune system alterations that persist postpartum. Studies have shown that parity can significantly alter the vaginal microbiome [38,43], potentially influencing susceptibility to viral persistence or reactivation. Thus, a potential explanation for the present observation, as previously suggested [10], is that quantitative alterations in the immune system occur as a consequence of pregnancy and that these changes result in an elevated likelihood of persistent TTV colonization. This finding that the detection of TTV in the circulation of pregnant women varies with parity is a pertinent variable that must be taken into consideration in investigations evaluating the presence of TTV as an immune marker of gestational disturbance.

Consistent with our findings, TTV has been detected in a substantial proportion of asymptomatic pregnant women who delivered at term [10]. However, unlike our plasma-based results, that study identified elevated concentrations of IL-6 and IL-8 in TTV positive vaginal samples, suggesting localized immune activation. These differences may reflect variation in sample type, anatomical site, or immune environment. The presence of TTV in vaginal fluid may indicate mucosal immune modulation, raising the possibility that TTV interacts differently with local and systemic immunity during pregnancy. Interestingly, even in a cohort that included preterm births, a prior study did not find an association between *Alphatorquetenovirus* detection in maternal blood and inflammatory complications such as preterm labor, histologic chorioamnionitis or preeclampsia [35]. Their findings, derived from a clinically vulnerable population, are consistent with ours and further suggest that the mere presence of TTV alone may not indicate a pathological state.

Limitations of the present study need to be acknowledged. The small number of study participants was a consequence of the relative rarity in our population of the availability for analysis of samples from each trimester in women with twin pregnancies. Additional studies on larger populations and from different geographical locations are needed to obtain more accurate results. Furthermore, there were no attempts to culture any of the herpesvirus positive samples, so we were unable to ascertain whether there was replicative virus present in any of the women. Another limitation is that pregnant women in our cohort were not routinely tested for antibodies to herpesviruses, alterations in the vaginal microbiome or for genital tract immune status. The only testing was for antibodies to rubella, hepatitis B virus s antigen, and hepatitis C virus. Thus, we could not determine the level of preexisting immunity to any of the viruses identified in the present study. It should be noted that none of the newborns had a fever the first few days after delivery. Moreover, our longitudinal design, with sampling in all three trimesters, allowed us to distinguish sustained from isolated virus detection – revealing a predominantly transient pattern without inflammatory correlates.

However, the detection of herpesviruses DNA in the circulation at one or two time points during gestation in women with twin pregnancies does not necessarily indicate the presence of an active infection nor predict the occurrence of a subsequent adverse pregnancy outcome. Similarly, TTV occurrence is common in women with uneventful twin pregnancies and its prevalence varies with parity, highlighting the importance of host-related factors in interpreting viral detection during pregnancy.

In conclusion, in twin pregnancies without adverse outcomes, the high frequency and variable viral titer of TTV and the transient circulation of herpesviruses in maternal plasma appear unaffected by the immunological adaptations inherent to this setting, as supported by the absence of correlation between inflammatory markers and viral dynamics. Nonetheless, ongoing viral monitoring of obstetric and associations with postnatal outcomes remain warranted.

## Supporting information

**S1 File. Supplementary information file.** This file contains all supplementary figures and tables.
(DOCX)

## Acknowledgments

We gratefully acknowledge the contributions of the researchers and staff from Careggi University Hospital, Florence, Italy, and the Virology Laboratory of the Institute of Tropical Medicine, Faculty of Medicine, University of São Paulo, Brazil, for their essential support and collaboration.

## Author contributions

**Conceptualization:** Maria Cassia Mendes-Correa, Steven S. Witkin.

**Formal analysis:** Tania Regina Tozetto-Mendoza, Layla Honorato, Maria Cassia Mendes-Correa, Steven S. Witkin.

**Investigation:** Tania Regina Tozetto-Mendoza, Mariane Pereira Brito, Noely Evangelista Ferreira, Angela Silvano, Antonio Charlys da Costa, Silvia Helena Lima, Viola Seravalli.

**Methodology:** Tania Regina Tozetto-Mendoza, Layla Honorato, Mariane Pereira Brito, Noely Evangelista Ferreira, Antonio Charlys da Costa, Silvia Helena Lima, Mariarosaria Di Tommaso.

**Project administration:** Maria Cassia Mendes-Correa.

**Supervision:** Maria Cassia Mendes-Correa, Steven S. Witkin.

**Writing – original draft:** Steven S. Witkin.

**Writing – review & editing:** Tania Regina Tozetto-Mendoza, Layla Honorato, Iara M. Linhares, Angela Silvano, Antonio Charlys da Costa, Mariarosaria Di Tommaso, Maria Cassia Mendes-Correa, Steven S. Witkin.

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
