## [Decision Letter · Decision Letter 0]

15 Jul 2025

Dear Dr. Tozetto-Mendoza,

Thank you for submitting your manuscript to PLOS ONE. After careful consideration, we feel that it has merit but does not fully meet PLOS ONE’s publication criteria as it currently stands. Therefore, we invite you to submit a revised version of the manuscript that addresses the points raised during the review process.

We look forward to receiving your revised manuscript.

Kind regards,

Simone Agostini, Ph.D.

Academic Editor

PLOS ONE

Journal Requirements:

The support for the laboratorial investigation was provided by the Laboratory of Medical Investigation (LIM) 52, Faculty of Medicine, University of São Paulo, Brazil.

5. Please remove all personal information, ensure that the data shared are in accordance with participant consent, and re-upload a fully anonymized data set.

Reviewers' comments:

Reviewer's Responses to Questions

**Comments to the Author**

1. Is the manuscript technically sound, and do the data support the conclusions?

Reviewer #1: No

Reviewer #2: Partly

Reviewer #3: Yes

2. Has the statistical analysis been performed appropriately and rigorously?

Reviewer #1: No

Reviewer #2: No

Reviewer #3: No

3. Have the authors made all data underlying the findings in their manuscript fully available?

Reviewer #1: Yes

Reviewer #2: Yes

Reviewer #3: Yes

4. Is the manuscript presented in an intelligible fashion and written in standard English?

Reviewer #1: Yes

Reviewer #2: Yes

Reviewer #3: Yes

Reviewer #1: Dear Authors,

Thank you for the opportunity to review your manuscript, which reports the detection of herpesviruses and Torquetenovirus (TTV) in plasma samples from 54 asymptomatic women with twin pregnancies, assessed in each trimester.

While the topic is of general interest, I found several major concerns:

1. The motivation for the study is unclear. Detection of latent herpesviruses and TTV in asymptomatic individuals is well known, and this manuscript does not test a specific biological hypothesis or provide novel insights beyond confirming known prevalence data.

2. The discussion is disproportionately long and unfocused. It extensively reviews prior literature without critically integrating your own findings or proposing clear interpretations or implications.

3. Results are reported mainly in tables with redundant text and no schematic or graphical summaries. The only significant association (TTV and multiparity) is mentioned but not meaningfully discussed.

4. Conclusions. The conclusion that viral detection lacks clinical significance is not adequately supported, as the study was not designed to assess clinical outcomes or biomarker utility.

Given these limitations, I believe the manuscript does not sufficiently advance understanding in the field in its current form.

Reviewer #2: Thank you for the opportunity to review this manuscript entitled "Detection of Herpesviruses and Torquetenovirus in Each Trimester in Asymptomatic Women Pregnant with Twins." The study presents an analysis of the presence of herpesviruses and Torquetenovirus across the three trimesters of pregnancy in a cohort of 54 asymptomatic women carrying twin pregnancies. The topic is moderately interesting, as it addresses a relatively understudied population and aims to contribute to our understanding of the virome dynamics during pregnancy. However, the sample size is relatively small, which limits the statistical power and generalizability of the findings. Moreover, the inclusion criteria are not clearly described, making it difficult to assess the representativeness of the cohort. The presentation of the results is also somewhat difficult to follow; a clearer structure and more detailed explanation of key findings would greatly improve the manuscript’s readability and impact. I have some suggestions that I believe could improve the clarity, methodological rigor, and overall impact of the manuscript.

Abstract:

The sentence “Herpesviruses and TTV were frequently detected” reflects the finding regarding TTV but no HSV, please reconsider.

Introduction

What is the main hypothesis of the study? What is the rationale for addressing TTV during pregnancy? Could this information be useful for clinical management at some point? Pregnancy is a particularly interesting period from an immunological perspective, and I agree that changes at the virome level may indeed occur. I encourage the authors to clearly formulate a hypothesis and elaborate on the potential significance of these viral changes. Introducing the study with a strong rationale will help readers understand why this research matters and why they should keep reading.

HSV can have deleterious effects on the fetus. Therefore, I am not convinced by the statement that “the clinical relevance of low-level viral DNA detection in plasma remains unclear.” While this may be true for viruses such as HHV-6 or TTV, in the case of HSV, VZV, and certainly CMV, reactivations during pregnancy can be clinically significant and even life-threatening for the fetus. I recommend that the authors revise this statement and acknowledge the potential risks associated with the reactivation of these specific viruses.

Are there data in singletons? Please summarize the evidence

Please introduce the cytokines you have chosen and explain why are you measuring precisely those ones

Methods

Were women tested for HSV, CMV, VZV … prior to inclusion? From my perspective they should all have positive serology for inclusion but it is unclear to me the design.

Were immunosuppressed women excluded? Please clarify as they may have higher TTV viral loads, for instance

Statistics: The section is very brief and the analysis seem pretty basic. Were VL transformed into logarithmic variables for the analysis? It is usually interesting to do so due to their distribution. Please clarify how were the data analyzed

Results

Line 130: All samples were negative for VSV. Do the authors mean VZV?

It is unclear to me if we refer to the same or different women in the sentence “Five women each were herpesvirus positive in the first or the second trimester

134 while four were positive in the third trimester.”

Please include data regarding age as this has been described in most papers as an important determinant of TTV presence (and it may interact with previous pregnancy)

The tables are very basic. Some Figures could be much more informative. Table 1 could include data regarding the positive women, were the same in each trimester or different? Maybe a graph with a spaghetti plot and a line for each positive participant with their viral loads? this would be more informative.

Table 4 could do so with TTV viral loads, per patient, with a median, for example, to see the evolution over time

Table 5 should include p values

Discussion

To my understanding, no immunological analysis were performed, so please refer to inflammation instead of “associated immune system activation”

Do we have data on HSV transmission? Follow up data on the neonates? As to say “Normal pregnancy progression and parturition in women positive for herpesviruses”

Can transient detection depend on the sample (small amount of blood maybe)

Taking into account the small numbers, I think authors should be more cautions when concluding “The transient detection of herpesviruses in maternal plasma during pregnancy—most commonly HHV-6 and HSV-1 in our cohort—was not associated with elevated proinflammatory cytokines or adverse obstetric outcomes.”

Line 185, please mention at least that on the other side, reactivation of some virus such as CMV during pregnancy associate important sequelae.

The Discussion section is overly focused on just one of the viruses analyzed (HSV-6), which leaves the impression that the other findings are of limited relevance. Given that multiple viruses were studied, the authors should aim for a more balanced discussion that addresses the potential significance of all the viral detections. This would provide a more comprehensive interpretation of the data and strengthen the overall contribution of the manuscript.

Once again, taking into account the numbers (4 women with HHV-6), please be more cautious with your conclusions “In our cohort, the absence of inflammation and adverse outcomes supports a non-pathogenic interpretation of HHV-6 DNA detection”

Line 214: be careful with CMV. Sometimes there are no sequelae, but others can infect the newborn and be life-threatening and we do not understand exactly why.

Line 221: according to some authors, you may expect a 100% prevalence of TTV. Please discuss your findings accordingly. Undetected? Not present?

How would you use TTV as a “potential marker of local immune status of pregnancy” What for?

It is not clear why the authors suggest a connection between blood TTV levels and the vaginal birome or local immununity, as TTV is ubiquitous. What is the rationale behind this assumption? Could the observed higher TTV loads simply reflect the maternal age of the participants? Age is known to influence immune status and TTV viral load, and it may act as a confounding factor in this analysis. I strongly recommend that the authors consider age as a potential confounder and include it in their statistical models. Have any multivariate analyses been performed to control for this and other possible confounding variables? Sex is also known to influence TTV viral load, with men having higher TTV viral loads. Have authors analyzed the sex of the babies? May hormones play a role here?

Please, include among limitations the small sample size (some virus only 1 woman!), reduces the statistical power to detect significant differences between groups. This limitation also affects the ability to perform robust multivariate analyses to adjust for potential confounders, such as maternal age or gestational age. As a result, some associations may have gone undetected or may be influenced by uncontrolled variables. A larger cohort would be necessary to strengthen the conclusions and allow for more comprehensive statistical modeling, ideally comparing to singletons. I am sure HIV was ruled put although it is not mentioned.

Conclusions

The conclusions, as currently written, are rather limited and do not fully reflect the potential implications of the findings. They lack depth and fail to provide a clear take-home message for the reader. I encourage the authors to expand this section by summarizing the key results, discussing their possible relevance in the context of pregnancy, and outlining directions for future research.

Reviewer #3: This study presents a novel longitudinal design to investigate the detection of herpesviruses and Torquetenovirus (TTV) in plasma samples of asymptomatic women pregnant with twins, across all three trimesters of gestation. Conducted in a well-defined cohort of 54 twin pregnancies, the research is unique in its trimester-specific sampling and simultaneous analysis of viral presence, inflammatory cytokine levels, and clinical pregnancy outcomes. While the results showed a high prevalence of TTV and transient herpesvirus detection, no association was found with systemic inflammation or adverse obstetric parameters. The focus on a typically underrepresented population, pregnant women with twin gestations, adds valuable insight to the field and strengthens the study’s relevance. However, some aspects warrant further clarification and discussion, as outlined below.

General comments:

There is increasing emphasis on incorporating a gender perspective in biomedical research. This study focuses on a highly relevant yet underrepresented population in research, pregnant women and newborns, which should be highlighted as a major strength in the discussion section. To further emphasize this aspect, I suggest referring to the participants as "women" rather than "patients" or "subjects" throughout the text.

In the main results and supplementary material, quantitative variables are presented inconsistently—some as means, others as medians. On what basis did the authors decide how to present each variable? Was a normality test performed to guide this choice? This should be explained in more detail in the Methods and Statistical Analysis sections.

-Abstract, titers were “quantified” instead of “detected”. When were the “levels of pro-inflammatory cytokines” determined?

-Introduction: onsider including and discussing the recently published work (DOI: 10.1038/s41598-024-73870-2) in the manuscript to enrich it with more up-to-date studies, as it establishes the association between TTV and the degree of immunosuppression, measured by the CD4/CD8 ratio, in HIV-infected patients receiving antiretroviral therapy.

-Methods: Specify the type of collection tube and the anticoagulant used for blood sampling, whether samples were collected under fasting conditions, as well as the subsequent processing steps for plasma isolation and its cryopreservation.

It is important to explain in the Methods section how values below the limits of detection were handled. In the case of TTV, its prevalence is considered to be 100% based on previous studies. For the analysis of TTV viral load, samples with undetectable TTV levels were assigned a value immediately below the lowest detected viral load to allow for log₁₀ transformation. Would the results change depending on how undetectable values are handled? In fact, patients with no detectable TTV in plasma did show a positive detection when analyzed in peripheral blood mononuclear cells (PBMCs).

The Statistical Analysis section should include the name and version of the statistical software used for data analysis.

-Results: For Table 1 and Table 3, please indicate the valid percentage for each count (i.e., percentages based on non-missing values). Additionally, clarify whether complete longitudinal data were available for all mothers across all time points.

In relation with soluble IL-6, previous observations showed a direct correlations with TTV levels in HIV infection (DOI: 10.1038/s41598-024-73870-2). Please, in the relevant section of the Discussion, indicate what might explain this lack of concordance or what result was expected.

For Table 5, please indicate which statistical tests were used to compare variables between groups, even if this information is already provided in the Methods section. Please specify the criteria used to define anemia in the study. Additionally, were there any significant differences in hemoglobin levels between the groups?

-Discussion: The manuscript discusses inflammation based on the measurement of only a limited number of soluble proinflammatory cytokines. It should be acknowledged as a study limitation that other important aspects of systemic immune activation and inflammation were not assessed. These include key immune components such as CD4+ T-cell counts, the CD4/CD8 ratio (a marker of immune progression in immunocompromised patients, which has shown strong correlations with TTV levels), as well as relevant metabolic parameters involved in immunometabolism during pregnancy, such as ferritin and hemoglobin levels.

**Do you want your identity to be public for this peer review?** For information about this choice, including consent withdrawal, please see our Privacy Policy

Reviewer #1: No

Reviewer #2: **Yes: ** Talia Sainz

Reviewer #3: No

---

## [Author Response · Author response to Decision Letter 1]

18 Sep 2025

Reviewer #1: Dear Authors,

Thank you for the opportunity to review your manuscript, which reports the detection of herpesviruses and Torquetenovirus (TTV) in plasma samples from 54 asymptomatic women with twin pregnancies, assessed in each trimester.

While the topic is of general interest, I found several major concerns:

The motivation for the study is unclear. Detection of latent herpesviruses and TTV in asymptomatic individuals is well known, and this manuscript does not test a specific biological hypothesis or provide novel insights beyond confirming known prevalence data.

Thank you for your comment. As we have more clearly included in the revised Abstract and Introduction in lines 22-24, 44-45, 65-74, this cohort is particularly relevant because twin gestations can impose distinct immunological challenges that may influence viral behavior.

We believe that by providing longitudinal data on this unique clinical population, our study offers insights that go beyond a simple confirmation of prevalence, as is now clarified in the Discussion section lines 216-220, 224-233, 244-224, 303-307).

The discussion is disproportionately long and unfocused. It extensively reviews prior literature without critically integrating your own findings or proposing clear interpretations or implications.

Answer: We have now revised the Discussion to make it more fluid and concise.

Results are reported mainly in tables with redundant text and no schematic or graphical summaries. The only significant association (TTV and multiparity) is mentioned but not meaningfully discussed.

Answer: we appreciate your observation. We now rephrased the sentences and recreated the Table 1 to clarify this point. We included the results of TTV and herpes in one single item (2.1) and a supplementary Figure 1S present the data on titer of detected viruses across all trimesters.

we would like to clarify that this point (TTV and multiparity) has been comprehensively discussed, beginning with an introductory sentence, in lines 259-270 of the Discussion section.

Conclusions. The conclusion that viral detection lacks clinical significance is not adequately supported, as the study was not designed to assess clinical outcomes or biomarker utility.

Given these limitations, I believe the manuscript does not sufficiently advance understanding in the field in its current form.

We agree that our study was not designed to evaluate the clinical utility of these viruses as biomarkers. We appreciate the opportunity to clarify this point, as we have done in lines 216-219and 304-308. We have restructured our conclusion to make it clear that the significance of our study lies not in assessing clinical outcomes, but in the longitudinal characterization of viral dynamics in this poorly studied population. This study characterizes viral dynamics that appear unaffected by the immunological adaptations inherent to twin pregnancies.

Reviewer #2: Thank you for the opportunity to review this manuscript entitled "Detection of Herpesviruses and Torquetenovirus in Each Trimester in Asymptomatic Women Pregnant with Twins." The study presents an analysis of the presence of herpesviruses and Torquetenovirus across the three trimesters of pregnancy in a cohort of 54 asymptomatic women carrying twin pregnancies. The topic is moderately interesting, as it addresses a relatively understudied population and aims to contribute to our understanding of the virome dynamics during pregnancy. However, the sample size is relatively small, which limits the statistical power and generalizability of the findings. Moreover, the inclusion criteria are not clearly described, making it difficult to assess the representativeness of the cohort. The presentation of the results is also somewhat difficult to follow; a clearer structure and more detailed explanation of key findings would greatly improve the manuscript’s readability and impact. I have some suggestions that I believe could improve the clarity, methodological rigor, and overall impact of the manuscript.

Thank you for your constructive comments. The sample size is a limitation of our study, as discussed in the first manuscript version and is now rephrased more concisely in lines 282-295. To date, a greater number of longitudinal studies with larger cohorts are still needed, especially at a time when the number of twin births is increasing, as justified now including the in lines 65-74 in Introduction section. The need for more research is underscored by the current demand for more data in the literature, given the scarcity of longitudinal studies on twin pregnancies. The inclusion criteria are now clarified in the Abstract section in lines 25-26, Materials and Methods section in lines [90-97]. To provide further clarity, we have also addressed this point in the Discussion (lines [216-217]) and throughout the manuscript.

Abstract:

The sentence “Herpesviruses and TTV were frequently detected” reflects the finding regarding TTV but no HSV, please reconsider.

Answer: we rephrased this sentence as lines 37-40.

Introduction

What is the main hypothesis of the study? What is the rationale for addressing TTV during pregnancy? Could this information be useful for clinical management at some point? Pregnancy is a particularly interesting period from an immunological perspective, and I agree that changes at the virome level may indeed occur. I encourage the authors to clearly formulate a hypothesis and elaborate on the potential significance of these viral changes. Introducing the study with a strong rationale will help readers understand why this research matters and why they should keep reading.

Answer: We added a new section to the Abstract (lines 22-24) and to the Introduction (lines 44-45; 53-55, 61-64) to clarify the study rationale and the relevance of assessing maternal immune status in twin pregnancies in relation to the TTV titer. As briefly justified in lines 75-79, this study provides the first longitudinal quantification of TTV and herpesvirus in maternal plasma and examines their inflammatory correlates, offering novel insights into viral dynamics across pregnancy trimesters.

HSV can have deleterious effects on the fetus. Therefore, I am not convinced by the statement that “the clinical relevance of low-level viral DNA detection in plasma remains unclear.” While this may be true for viruses such as HHV-6 or TTV, in the case of HSV, VZV, and certainly CMV, reactivations during pregnancy can be clinically significant and even life-threatening for the fetus. I recommend that the authors revise this statement and acknowledge the potential risks associated with the reactivation of these specific viruses.

Thank you for your valuable feedback. Undoubtable, we agree that the detection of HSV, VZV, and CMV DNA in maternal plasma has important clinical implications due to potential fetal risks. The term 'unclear' referred specifically to the difficulty in unequivocally distinguishing latent from lytic infection phases based solely on plasma DNA detection, in contrast to whole blood, where latent viral DNA may be present in cellular compartments. We have now clarified this point in lines 221-230 in the Discussion section.

Are there data in singletons? Please summarize the evidence

Many studies have reported that TTV is quite common in women with singleton pregnancy outcomes, representing a possible predictor of local immune status, preterm birth risk and labor-related events, stated in lines 251-253 and we added sentences to summarize such evidence in lines 51-55 in the Introduction section.

Please introduce the cytokines you have chosen and explain why are you measuring precisely those ones

We have clarified this point by rephrasing lines 106–108 in the Materials and Methods section: interleukin (IL)-1β, IL-6, and tumor necrosis factor-α in these women had been previously described in twin pregnancies and were shown to significantly increase across the trimesters of gestation".

Methods

Were women tested for HSV, CMV, VZV … prior to inclusion? From my perspective they should all have positive serology for inclusion but it is unclear to me the design.

Were immunosuppressed women excluded? Please clarify as they may have higher TTV viral loads, for instance

We agree that women were not routinely tested for herpes serology prior to inclusion, and this limitation is acknowledged in the Discussion section (lines 288–289). We now clarified this point in Materials and methods in lines 86-91.

Statistics: The section is very brief and the analysis seem pretty basic. Were VL transformed into logarithmic variables for the analysis? It is usually interesting to do so due to their distribution. Please clarify how were the data analyzed

Thank you for your observation. We have now clarified this point in lines 140-142. We transformed into logarithmic the viral load (titer), according data in the lines 146-147.

Results

Line 130: All samples were negative for VSV. Do the authors mean VZV?

We apologize for the typo on Line 160. The reviewer is correct; we indeed meant VZV (Varicella-Zoster Virus), not VSV. We have corrected this typo in the manuscript.

It is unclear to me if we refer to the same or different women in the sentence “Five women each were herpesvirus positive in the first or the second trimester while four were positive in the third trimester.”

Thank you for your observation. We revised the Table 1 to clarify this point.

Please include data regarding age as this has been described in most papers as an important determinant of TTV presence (and it may interact with previous pregnancy)

Thank you for your observation. We now added such data in lines 96-97: the median age at sampling was 35 years (IQR: 33–38), with values ranging from 26 to 46 years.

The tables are very basic. Some Figures could be much more informative. Table 1 could include data regarding the positive women, were the same in each trimester or different? Maybe a graph with a spaghetti plot and a line for each positive participant with their viral loads? this would be more informative.

We appreciate your suggestion and have revised Table 1 and included Supplementary Tables S2, S3, and Figure S1.

Table 4 could do so with TTV viral loads, per patient, with a median, for example, to see the evolution over time. Table 5 should include p values.

We have reviewed and compiled the data in one single Table 1 and in the Supplementary Tables (S2, S3 and Figure 1S)

Discussion

To my understanding, no immunological analysis were performed, so please refer to inflammation instead of “associated immune system activation”

We have now rephrased the sentence to present the information more directly, assessing the correlation between pro-inflammatory cytokines level and viral titer throughout the manuscript.

Do we have data on HSV transmission? Follow up data on the neonates? As to say “Normal pregnancy progression and parturition in women positive for herpesviruses” Can transient detection depend on the sample (small amount of blood maybe) Taking into account the small numbers, I think authors should be more cautions when concluding “The transient detection of herpesviruses in maternal plasma during pregnancy—most commonly HHV-6 and HSV-1 in our cohort—was not associated with elevated proinflammatory cytokines or adverse obstetric outcomes.”

Thank you for your observation. We have clarified this point, noting that the study analysis was conducted in healthy women who delivered at term without complications (lines 90–96, Materials and Methods section). In the Discussion section, we have added a note of caution regarding the risk of adverse events (lines 228–231) as well as in the conclusion (303-307) and limitation of this study (282-287).

Line 185, please mention at least that on the other side, reactivation of some virus such as CMV during pregnancy associate important sequelae.

we thank the reviewer for this valuable suggestion. We have revised the manuscript to acknowledge that, in contrast, the reactivation of certain viruses such as CMV during pregnancy may be associated with significant sequelae. This addition has been made in line 232-234 of the Discussion section.

The Discussion section is overly focused on just one of the viruses analyzed (HSV-6), which leaves the impression that the other findings are of limited relevance. Given that multiple viruses were studied, the authors should aim for a more balanced discussion that addresses the potential significance of all the viral detections. This would provide a more comprehensive interpretation of the data and strengthen the overall contribution of the manuscript.

Once again, taking into account the numbers (4 women with HHV-6), please be more cautious with your conclusions “In our cohort, the absence of inflammation and adverse outcomes supports a non-pathogenic interpretation of HHV-6 DNA detection”

Line 214: be careful with CMV. Sometimes there are no sequelae, but others can infect the newborn and be life-threatening and we do not understand exactly why.

Line 221: according to some authors, you may expect a 100% prevalence of TTV. Please discuss your findings accordingly. Undetected? Not present?

How would you use TTV as a “potential marker of local immune status of pregnancy” What for?

It is not clear why the authors suggest a connection between blood TTV levels and the vaginal virome or local immununity, as TTV is ubiquitous. What is the rationale behind this assumption? Could the observed higher TTV loads simply reflect the maternal age of the participants? Age is known to influence immune status and TTV viral load, and it may act as a confounding factor in this analysis. I strongly recommend that the authors consider age as a potential confounder and include it in their statistical models. Have any multivariate analyses been performed to control for this and other possible confounding variables? Sex is also known to influence TTV viral load, with men having higher TTV viral loads. Have authors analyzed the sex of the babies? May hormones play a role here?

Please, include among limitations the small sample size (some virus only 1 woman!), reduces the statistical power to detect significant differences between groups. This limitation also affects the ability to perform robust multivariate analyses to adjust for potential confounders, such as maternal age or gestational age. As a result, some associations may have gone undetected or may be influenced by uncontrolled variables. A larger cohort would be necessary to strengthen the conclusions and allow for more comprehensive statistical modeling, ideally comparing to singletons. I am sure HIV was ruled put although it is not mentioned.

Discussion focus on HSV-6:

We thank the reviewer for this comment. We have revised the Discussion to provide a more balanced interpretation of all viral detections, highlighting the potential relevance of more frequent herpes detected.

Caution with HHV-6 conclusions (4 women):

We agree and have revised the conclusions to reflect caution, acknowledging the small number of HHV-6 cases and the limited statistical power. We clarified that HHV-6 is no uncommon in pregnancy, and we cannot exclude the risk of reactivation, including chromosomally integrated HHV-6, as we now indicated in lines 242-244.

CMV

We have added a note emphasizing that CMV reactivation can sometimes lead to severe congenital, perinatal and neonatal infection, while other cases may remain asymptomatic, as revised now in lines 232-234 and lines 229-230.

TTV prevalence and TTV marker

We have clarified our findings regarding TTV prevalence, discussing them in the context of previously reported prevalence in both the Introduction (lines 51–54) and Discussion sections (lines 246–252). Importantly, studies from our group have demonstrated associations between TTV, vaginal dysbiosis, and the local inflammatory microenvironment, reinforcing its potential role as a marker reflecting local immune status through TTV titer analysis.

Confounding factors (age, sex, hormones):

We thank the reviewer for raising this point. In our cohort, maternal age did not show a significant correlation with TTV viral load across the three trimesters (Spearman r = -0.021, 0.020, and -0.

---

## [Decision Letter · Decision Letter 1]

8 Oct 2025

Circulation of Herpesvirus and Alphatorquevirus DNA in Each Trimester in Asymptomatic Women Pregnant with Twins

PONE-D-25-31241R1

Dear Dr. Tozetto-Mendoza,

We’re pleased to inform you that your manuscript has been judged scientifically suitable for publication and will be formally accepted for publication once it meets all outstanding technical requirements.

Kind regards,

Simone Agostini, Ph.D.

Academic Editor

PLOS ONE

Additional Editor Comments (optional):

Reviewers' comments:

Reviewer's Responses to Questions

**Comments to the Author**

Reviewer #2: All comments have been addressed

Reviewer #3: All comments have been addressed

2. Is the manuscript technically sound, and do the data support the conclusions?

Reviewer #2: Yes

Reviewer #3: Yes

3. Has the statistical analysis been performed appropriately and rigorously?

Reviewer #2: Yes

Reviewer #3: Yes

4. Have the authors made all data underlying the findings in their manuscript fully available?

Reviewer #2: Yes

Reviewer #3: Yes

5. Is the manuscript presented in an intelligible fashion and written in standard English?

Reviewer #2: Yes

Reviewer #3: Yes

Reviewer #2: Great work! We gave you trouble but I think the paper clearly improved after the revision. Congratulations!

Reviewer #3: I have carefully evaluated the revised version of the manuscript entitled “Circulation of Herpesvirus and Alphatorquevirus DNA in Each Trimester in Asymptomatic Women Pregnant with Twins” (PONE-D-25-31241R1).

The authors have adequately addressed the previous comments, and the changes introduced have significantly improved the clarity and quality of the manuscript. I consider the revisions satisfactory, and in my opinion, the article is now ready for publication in PLOS ONE.

**Do you want your identity to be public for this peer review?** For information about this choice, including consent withdrawal, please see our Privacy Policy

Reviewer #2: No

Reviewer #3: No

---

## [Editor Report · Acceptance letter]

PONE-D-25-31241R1

PLOS ONE

Dear Dr. Tozetto-Mendoza,

I'm pleased to inform you that your manuscript has been deemed suitable for publication in PLOS ONE. Congratulations! Your manuscript is now being handed over to our production team.

Kind regards,

on behalf of

Dr. Simone Agostini

Academic Editor

PLOS ONE